



# Assessing multivariate bias corrections of climate simulations on various impact models under climate change

Denis Allard[1], Mathieu Vrac[2], Bastien François[3], and Iñaki García de Cortázar-Atauri[4]

[1]Biostatistisc and Spatial Processes (BioSP), INRAE, Avignon 84914,France
[2]Laboratoire des Sciences du Climat et de l'Environnement (LSCE-IPSL), CEA/CNRS/UVSQ, Université Paris-Saclay, Centre d'Etudes de Saclay, Orme des Merisiers, 91191 Gif-sur-Yvette, France
[3]Royal Netherlands Meteorological Institute (KNMI), Research and Development Weather and Climate (RDWK), De Bilt, The Netherlands
[4]US Agroclim, INRAE, Avignon 84914, France

**Correspondence:** Denis Allard (denis.allard@inrae.fr)

**Abstract.** Atmospheric variables simulated from climate models often present biases relative to the same variables calculated by reanalysis in the past. In order to use these models to assess the impact of climate change on processes of interest, it is necessary to correct these biases. Currently, the bias correction methods used operationally correct one-dimensional time series and are therefore applied separately, physical variable by physical variable and site by site. Multivariate bias correction methods

have been developed to better take into account dependencies between variables and in space. Although the performance of multivariate bias correction methods for adjusting the statistical properties of simulated climate variables has already been evaluated, their effectiveness for different impact models has been little investigated. In this work, we propose a comparison between two multivariate bias correction methods (R2D2 and dOTC) in three different configurations (intervariable, spatial and spatial-intervariable) and a univariate correction (CDF-t) through several highly multivariate impact models (phenological

stage, reference evapotranspiration, soil water content, fire weather index) integrating the weather conditions over a whole season. Our results show that CDF-t does a fair job in most situations and that there is no single best MBC method. The performances of multivariate bias correction methods depend both on some characteristics of the studied process and on the configuration of the chosen bias correction method. When all characteristics are important (multivariate, time cumulative and spatial) it is found that dOTC in its spatial-intervariable configuration brings improvements in most cases and no significant

improvement in some rare cases. We did not find any multivariate cases where the spatial-intervariable configuration for dOTC performs less well than CDF-t.

**Keywords.** Bias adjustment, Climate model, Agroclimatic indicators, Evapotranspiration, Pheonology model, Fire Weather Index, Soil Water Content, Spatial analysis.

## 1 Introduction

In the context of climate change, anticipating and implementing mitigation and adaptation strategies using climate model simulations is crucial to understand the possible consequences of future climate on human societies. These projections are





generated by global and regional climate models (GCMs and RCMs) that are based on well-established physical principles and constrained by several greenhouse gas emission scenarios, prescribed for example as part of the CMIP6 project (Eyring et al., 2016). By using these projections as input into impact models, consequences of climate change can be anticipated in a various number of domains, for instance to investigate changes of vegetation distribution (e.g., Bachelet et al., 2001; Zhang et al., 2023; Chuine, 2010), agricultural production (e.g., Bezner Kerr et al., 2022; Wheeler and von Braun, 2013; Zhu et al., 2022), global water resources (e.g., Bates et al., 2008; Hagemann et al., 2013), biodiversity (Bellard et al., 2012), spread of epidemic diseases (e.g., Caminade et al., 2014; Chemison et al., 2021).

Hence, providing reliable climate information is essential to obtain robust impact assessments. However, despite considerable scientific progress in climate modeling, climate simulations often present biases compared to observations. This means that, even over the historical period, key statistical properties such as mean, variance or spatial correlations between physical variables may differ from those observed (see, e.g., Christensen et al., 2008; Eden et al., 2012; Cattiaux et al., 2013; Mueller and Seneviratne, 2014). Climate projections for future periods can therefore also be expected to be biased, making some form of bias adjustment necessary before they can be used as inputs into impact models (Maraun et al., 2010; Teutschbein and Seibert, 2012). To alleviate such errors, many statistical bias correction (BC) methods have been developed over the last decades and aim to produce "adjusted" climate simulations. Statistical bias correction consists in transforming climate model simulations to align (a selected set of) their statistical features with those of a reference data set over the historical period. Then, this transformation can be applied to future simulations to obtain adjusted outputs for the projection period. Simple univariate statistical features of climate variables can be targeted for correction, such as the mean (using delta-change, e.g., Xu, 1999) or the variance (using scaling of variance, e.g., Schmidli et al., 2006; Eden et al., 2012; Berg et al., 2012). In general, the univariate approaches the most commonly applied rely on the quantile-mapping technique (e.g., Haddad and Rosenfeld, 1997; Déqué, 2007; Michelangeli et al., 2009; Gudmundsson et al., 2012; Vrac et al., 2012; Cannon et al., 2015) that adjust not only the simulated mean and variance but also all percentiles. However, by adjusting simulated variables separately for each physical variable at each specific location, univariate bias correction methods are unable to adjust potential biases in the simulated inter-variable or spatial properties (e.g., correlations). Studies show that independent applications of univariate bias correction methods do not modify the inter-variable or inter-site dependence structure of the simulated variables to be corrected (e.g., Wilcke et al., 2013; Ivanov and Kotlarski, 2017; Vrac, 2018), which can therefore lead to inappropriate multivariate situations if these dependencies are not correctly represented in the climate model. This can have significant consequences when these corrections are used as inputs into impact models that rely on non-linear relationships between multiple climate variables at various spatial and temporal scales. Indeed, if the statistical dependencies between input climate variables are not realistically simulated, then biases can propagate to simulated impacts that depend on multivariate interactions, regardless of whether simulations are corrected using univariate BC methods (e.g., Boé et al., 2007; Zscheischler et al., 2019). A correct representation of climatic variables and their dependencies is thus necessary for many impact studies: for instance, appropriate inter-variable and spatial properties are of paramount importance for spatio-temporal wildfire risk assessments that are determined by complex interactions between wind, temperature, relative humidity and precipitation (Senande-Rivera et al., 2022; Barik and Baidya Roy,





2023), or for flood risk assessments, for which the spatial (and temporal) properties of precipitation, soil moisture and water are involved (Vorogushyn et al., 2018).

To adjust biases in multivariate dependencies, some multivariate bias correction (MBC) methods have been developed in the literature. The objective of MBC is fundamentally the same as that of univariate BC: transforming climate model simulations so that selected statistical features match those of a reference data set over the calibration period. The difference with univariate BCs lies in the fact that statistical features are not only univariate, but also multivariate such as inter-variable correlations or spatial copula structure. MBC methods can be grouped into three categories based on how they adjust climate simulations (Vrac, 2018; Robin et al., 2019; François et al., 2020): 1) the "marginal/dependence" category that gathers MBCs adjusting separately univariate distributions and multivariate properties (e.g., Bárdossy and Pegram, 2012; Mehrotra and Sharma, 2016; Vrac, 2018; Nahar et al., 2018; Cannon, 2018; Nguyen et al., 2019; Guo et al., 2019; Vrac and Thao, 2020; François et al., 2021); 2) the "successive conditional" correction approach, that adjusts the different simulated variables successively and conditionally on the previously adjusted ones (e.g., Piani and Haerter, 2012; Dekens et al., 2017); 3) the "all-in-one" category that consists of MBCs simultaneously adjusting univariate and multivariate properties of climate simulations (e.g., Robin et al., 2019; Pan et al., 2021).

Due to the differences in the applicability of the various MBC methods, as well as their various underlying assumptions and statistical methods used, the quality of multivariate bias-adjusted outputs can differ (François et al., 2020; Guo et al., 2020). François et al. (2020) carried out an intercomparison study of four MBC methods to adjust simulated temperature and precipitation outputs. Results show that half of the methods were able to reasonably correct simulated inter-variable and spatial dependence structures, while some of them presented instability issues. However, even if multivariate bias-adjusted outputs have appropriate statistical properties, differences of quality can potentially transfer into the often non-linear impact model outputs. More generally, the uncertainty introduced by bias correction in the impact modeling chain, in addition to the other sources of uncertainty (e.g., choice of climate models, forcing scenarios and impact models), requires to be explored on a case-by-case basis (Rötter et al., 2012; Tao et al., 2018). Regarding univariate BC methods, recent studies have demonstrated their effectiveness for specific regional impact studies, for instance for hydrological purposes (e.g., Teutschbein and Seibert, 2012; Chen et al., 2013; Hakala et al., 2018), forest fire prevention (e.g., Yang et al., 2015; Casanueva et al., 2018) or agriculture applications (e.g., Oettli et al., 2011; Laux et al., 2021). However, for multivariate BC methods, their suitability for impact studies is the subject of debates within the scientific community. While, for a specific hydrological application, Räty et al. (2018) found that using MBC methods is not necessarily beneficial compared to using less sophisticated univariate BC methods, other studies demonstrated their added value, for instance to improve the realism of simulated multivariate fire weather indices (e.g., FWI, Cannon, 2018; Casanueva et al., 2018; Zscheischler et al., 2019), multivariate drought indices (Adeyeri et al., 2023; Ansari et al., 2023), simulated carbon cycle (Teckentrup et al., 2023), impact on crop modeling (Galmarini et al., 2024) or hydrological impact projections (Chen et al., 2018; Meyer et al., 2019; Singh and Reza Najafi, 2020; Su et al., 2020; Tootoonchi et al., 2022; Vogel et al., 2023), although these benefits being less pronounced in non-stationary contexts (Guo et al., 2020; Van de Velde et al., 2022). These conflicting results could be potentially explained by the fact that these studies sometimes apply a set of methods in a limited number of dimensional configurations that does not allow the BC-induced





uncertainty to be adequately covered. Also, these studies often use impact indicators that depend on statistical features of climate variables that are not necessarily adjusted by MBC methods (e.g., temporal properties). Identifying all the statistical characteristics involved in impact metrics is an important aspect to better understand the outputs from impact models and to provide the nuances needed to correctly assess performances of MBC methods. In addition, a large majority of these studies applies MBCs to adjust simulated univariate and inter-variable properties in low dimensional contexts and discarding spatial consideration (with the exception of Ahn et al., 2023) while the accurate representation of spatial dependence is also relevant to many impact studies. A more comprehensive overview of the effectiveness of the MBC methods for impact models is therefore needed, not only by considering several MBC methods and different impact metrics, but also by assessing their capacity to provide reliable spatial information that is essential for impact studies.

In this study, which can be seen as a follow-up of François et al. (2020), we present an analysis of two multivariate bias correction methods applied to adjust the inter-variable and/or spatial dependence structures of 5 physical variables (daily mean temperature, total precipitation, near-surface wind speed, short-wave downwelling radiation and near-surface relative humidity). This study thus complements the analysis on crop models in Galmarini et al. (2024) which showed the added value of multivariate bias correction methods on yield but did not include a spatial dimension, as well as the findings reported in Ahn et al. (2023) which focused on hydrological applications using only three climate variables (daily precipitation, minimum and maximum daily temperatures).

We focus in particular our analysis on MBC methods that present stable results in high-dimensional contexts, according to François et al. (2020). We evaluate their performances using four impact metrics (phenological stage, evapotranspiration, soil water content and fire weather index) from agronomic and forest impact models for three subregions of France in order to better understand the influence of MBCs. A univariate BC method is also included in the study to assess the potential benefits of considering multivariate aspects. In addition to providing a diversified intercomparison framework, the three subregions were chosen to provide relevant adjusted data to climate services that can be reused in the scientific community. This permits to have an extensive overview of the performance of the multivariate bias correction methods for impact studies and further identifying their advantages and limits.

This paper is organized as follows: Section 2 describes the climate model and reference data, the agronomic and forest impact models and the multivariate correction methods. The experiment setup and the statistical analyses — focusing on spatial features — are presented in Sect. 3, as well as the notion of Effective Sample Size, central to the hypothesis testing in a spatio-temporal context. Section 4 presents key selected results among those obtained. Finally, our findings are summarized in Sect. 5, along with guidelines for users, element of discussions and perspectives for future research.

## 2    Data, models and bias correction methods

### 2.1    Model simulations and reference data

The climate model used in this study is the IPSL-CM6A-LR coupled model (Boucher et al., 2020) developed at the Institut Pierre-Simon Laplace (IPSL), part of the 6[th] Coupled Models Intercomparison Project (CMIP6, Eyring et al., 2016). Daily





values of 5 physical variables that are used as input variables for impact models have been extracted over a historical period

(1985-2014), which will be used for comparison and calibration, and a future period (2036-2065): daily mean temperature (tas), total precipitation (pr), near-surface wind speed (scfWind), short-wave downwelling radiation (rsds) and near-surface relative humidity (hurs). We selected the ssp585 (SSP5-RCP8.5) scenario, i.e. the scenario with the highest $CO_2$ concentration.

Since our study covers France, the reference data is the gridded "Systeme d'Analyze Fournissant des Renseignements Atmosphériques à la Neige" (SAFRAN) reanalysis dataset (Vidal et al., 2010). Daily time series of the same 5 variables have a

130 8 km × 8 km spatial resolution and divide France into 8981 contiguous continental grid cells. IPSL-CM6A-LR data, available at the 2.5° × 1.3° resolution, were regridded to the SAFRAN resolution using the nearest-neighbor technique.

Since the historical reference data SAFRAN is available for the same period (1985-2014), a statistical comparison between bias corrected time series and reference is possible. Then, during the future period, comparison between the projection and the historical periods are possible for each bias correction method. Moreover, comparison between bias corrected projections is

135 also possible.

Three contrasted regions of France were selected, see Fig. 1): Brittany (North-West part of France, 259 grid cells), Ile-de-France (the region around Paris, 319 grid cells) and Provence (South-East part of France, 337 grid cells). In the latter, some grid cells are located in the Alps, with quite high mean elevation, up to 2900 m. In these grid cells the temperatures are significantly lower and the precipitations are significantly higher than in other locations of the region.

## 2.2 Impact models

The physical variables described above are used as input variables for several process models in order to compute indicators. Four impact models have been selected based on their characteristics, which are listed in Table 1 with their main characteristics. They are briefly discussed here before being detailed in the rest of this section. The widely used reference evapotranspiration, ET0, is computed on a daily basis. Different phenological plant models have been used to describe main phenological stages,

which depend only on Temperature. They provide one date for each phenological stage per season. Here, we will use flowering stage (FLO). Then, using a complete plant model, a cumulative water balance is computed every day. We will use as indicator the Soil Water Content (SWC). Finally, the Fire Weather Index (FWI), which is a danger indicator for forest wildfires, is also computed. In summary, FLO depends only on temperature and is cumulative. SWC, ET0 and FWI return daily values involving all physical variables, except ET0 which does not include Precipitation. All indicators are cumulative, except ET0.

| | Acronym | Time integration | Multivariate | Daily |
|---|---|---|---|---|
| Evapotranspiration | ET0 | × | ✓ | ✓ |
| Phenological stage | FLO | ✓ | × | × |
| Soil Water Content | SWC | ✓ | ✓ | ✓ |
| Fire Weather Index | FWI | ✓ | ✓ | ✓ |

**Table 1.** Main features of the different impact model indicators



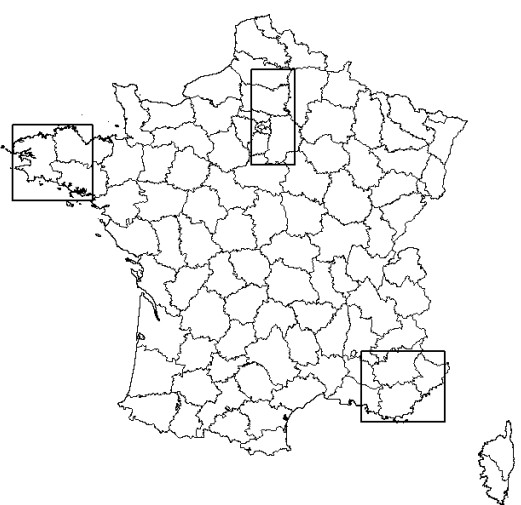

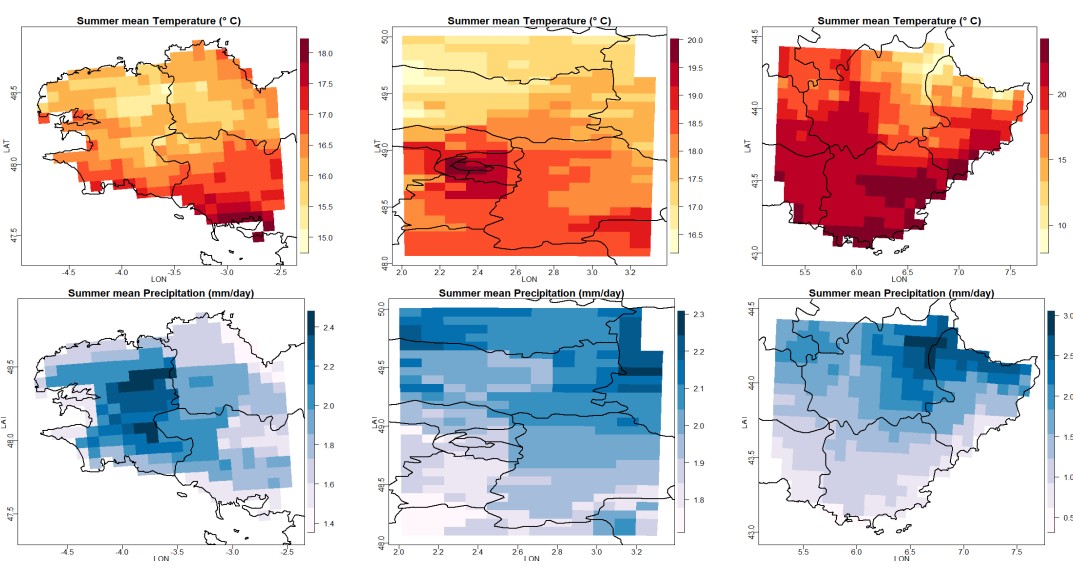

**Figure 1.** Top: map of France showing the three studied regions in France: Brittany (North-West), Ile-de-France (North-center) and Provence (South-East). Middle and Bottom: summer (JJA) mean temperature (middle) and summer mean daily precipitation (bottom) using SAFRAN database over the three studied regions and the 1985 - 2014 period : Brittany (left), Ile-de-France (middle) and Provence (right). In Ile-de-France, the Paris hot spot is visible in the middle of domain (top, center). In Provence, the influence of the Alps, with lower temperatures (top, right) and higher precipitations (bottom right) than elsewhere is visible in the upper right corner



**Reference evapotranspiration**

The reference evapotranspiration, ET0 (in mm), is a classical and very informative indicator allowing to describe the plant-soil water losses during the day. It is computed directly from all variables except precipitation, using the Penman–Monteith formula (Allen et al., 1998), independently to any chosen plant model. ET0 is computed every day of the year, separately at each grid cell. As indicated in Table 1, ET0 depends thus on marginal and inter-variable properties, with no time integration.

**Flowering**

Phenology is considered as the first biophysical indicator of climate change (Menzel et al., 2006) and it is currently used to explore crops climate suitability (Caubel et al., 2015). We propose to use simple phenological models to simulate main phenological stages for different crops. Most of these models depend on the accumulation of temperature only (Chuine et al., 2013). Three plant models, among the major crops cultivated in France, have been selected: wheat, maize and grapevine.

– Wheat is mostly cultivated in and around Ile-de-France, but wheat fields can be found anywhere in France, except at high elevations. Flowering stage corresponds to a sum of positive temperature equal to 375 taking into account previous phenological stages which also depend on photoperiod (not affected by the methods described below). Sowing date was set to October 1st, every year and for all grid cells.

– A short-cycle variety was chosen for maize in order to achieve late phenological stages in the Northern regions (Brittany and Ile-de-France). Flowering stage corresponds to a sum of effective temperature (above 6°) equal to 120 from the sowing date. Since no irrigation was introduced, water deficit can be expected in the Southern region (Provence). Sowing date was set to April 10, every year and for all grid cells.

– Vineyards are very common in Provence (except in the Alps), possible in the Ile-de-France region (e.g. nearby Champagne) and they are being recently developed in Brittany. A rather ubiquitous variety, Chardonay, was chosen. There is no sowing date for grapevines, but the whole model was re-initialized August 1st of the previous year and calculated intermediate phenological stages (as dormancy and budbreak).

We calculated the flowering stage (FLO) for all the three species because it is an important phenological stage which allow to characterize crop potential production. Moreover, flowering is the latest stage reached in most grid cells and years, except perhaps in the grid cells covering high mountains in the Provence domain. For one given year, FLO is thus the date, expressed in Julian days. The interest of these phenological models is that depend thus only on univariate properties and it is time cumulative. Global framework and models are described in (Caubel et al., 2015), see also Maury et al. (2021) and Garcia De Cortazar Atauri and Maury (2019). All the phenological models were translated into R scripts.

**Water balance model**

We also calculated the Soil Water Content (SWC) using a simplified water balance model (Allen et al., 1998). This model combines the reference evapotranspiration (ET0 - above described) and the daily precipitation over the time. The evolution of





the amount of water available in the soil also takes into account the type of crop planted in the soil (using ks coefficient). SWC is the daily water content in the soil, expressed as a percentage of the total quantity available (the 'Water Reserve'), taking into account a soil description. SWC involves all 5 variables and it has a strong temporal component relating to the temporal occurrence of precipitation. With this indicator, it is possible to assess whether or not the MBCs are able to correct a highly

non-linear transformation of the 5 variables. In comparison to ET0, since SWC is cumulative, it will be possible to assess the effect of the MBCs on time-correlated outputs.

In order to avoid problems of soil heterogeneity in each region and to catch as much as possible climate effect, we used a single type of soil. Thus, a deep soil (140 cm), with more than 200 mm useful water reserve and moderate soil water capacity (28% at field capacity and 35% at saturation) was chosen for all grid cells in all regions.

**Fire Weather Index**

With nearly one third of Metropolitan France (mainland France and Corsica, without overseas territories) covered by woods and forests, France has among Europe's highest forest cover (MTES, 2021). Apart from providing resources and recreational activities, the forest plays a key role in climate regulation, the water cycle, and soil preservation including its role as biodiversity reservoir, carbon sink and in erosion control. However, weather conditions such as drought, temperature and wind have strong

influence on the forests' vulnerability to fire and its potential for spreading. Between 2007 and 2019, wildfires destroyed around 11,500 ha of forest per year in Metropolitan France. In 2022, over 59,000 ha of forests were destroyed (https://bdiff.agriculture. gouv.fr/incendies). Climate change increases the weather-induced component of the forest fire risk across France and Europe (Dupuy et al., 2020; Fargeon et al., 2020; Ruffault et al., 2020). The Fire Weather Index (Van Wagner et al., 1987) summarizes the effects of wind, temperature, humidity and precipitation into a single index used by the national security services as a

danger rating system for forest wildfires. In theory, the FWI is determined every day from the FWI value of the preceding day and from noon weather readings: temperature, relative humidity, wind speed, and rain (if any). Similarly to SWC, FWI depends thus on marginal, inter-variable and temporal properties, the latter not being accounted for in the tested correction methods of this study. In this work, FWI is daily computed based on simulation or reanalysis data instead of meteorological readings, using the R package `cffdrs` (Wang et al., 2017).

**2.3 Multivariate bias correction methods**

Multivariate Bias Correction methods (MBC) must be chosen for their capacity to correct the biases and their spatial properties over large geographical areas. Following the discussion in François et al. (2020), good candidates are dOTC (dynamical Optimal Transport Correction) and R2D2 (Rank Resampling for Distributions and Dependencies). These methods are briefly recalled here and we refer to François et al. (2020) and the original papers referenced therein for a more in-depth presentation.

R2D2, proposed in Vrac (2018) consists in two steps. In the first step, each climate variable is adjusted using a univariate bias correction method. In this work, CDF-t (described below) is used, but other methods could be used. The second step is essentially a re-ordering technique, called the Schaake Shuffle (Clark et al., 2004) which reorders a sample such that the rank structure corresponds to the rank structure of a reference sample. A reference dimension (i.e. one physical variable at one given





site) is selected, for which the rank chronology of the simulations remains unchanged. Reconstruction of the inter-variable and

spatial rank correlations of the reference is then performed, while preserving the rank temporal dynamics for the reference dimension. It must be emphasized that, by construction, R2D2 assumes the intervariable and spatial rank correlations to be stationary in time.

dOTC (Robin et al., 2019) corrects the marginal distributions and the multivariate dependence at the same time. It is a generalization of the univariate quantile mapping approach to the multivariate case. Based on optimal transport theory, it builds

a transfer plan, which is a multivariate transfer function from one multivariate distribution to another, that minimizes a cost function based on an energy criterion. Two important differences between R2D2 and dOTC are that dOTC does not single out a particular "reference dimension" and that dOTC does not assume the stationarity of copula structure between the calibration and the projection periods. dOTC is designed to transfer some of the multivariate properties' changes between the calibration and the projection periods from the model to bias corrected data.

As a benchmark, the multivariate dataset is also corrected using the univariate CDF-t correction method (Michelangeli et al., 2009). Separately for each variable and at each site, CDF-t estimates a univariate transfer function, denoted $T$, that links the Cumulative Distribution Function (CDF) of a climate variable of interest in the model simulations during calibration period to that of the same variable in the reference dataset. By assuming that $T$ is also valid during the projection period, a "future reference" CDF can be defined, by applying $T$ to the future climate model CDF. Then, a quantile-quantile approach is performed

between the new reference CDF and the CDF from the model simulations during the projection period. CDF-t is designed to take into account potential simulated changes (between calibration and projection periods) of the univariate distribution in the correction procedure. Thus, the bias-corrected data for the projection period incorporate the model's projected changes. In the specific case of precipitations, the "Singularity Stochastic Removal" version of CDF-t (Vrac et al., 2016) is applied, working the same way as CDF-t but specifically designed to account for rainfall occurrences.

## 3 Experiments and statistical analysis

### 3.1 Experiments

All indicators (ET0, FLO, SWC, FWI) are computed using the 5 physical variables described in Sect. 2.1 (tas, pr, scfWind, rsds and hurs) simulated by the IPSL model gridded to the SAFRAN 8 km × 8 km mesh (hence, with no correction) or corrected on the same grid using one of the bias correction methods: CDF-t, dOTC and R2D2. The MBC methods dOTC and R2D2 are

applied according to the three following configurations:

- The Intervar configuration (I) aims at correcting inter-variable correlations only: the MBC method corrects jointly the 5 physical variables at each grid cell of the domain independently on all other grid cells. In this configuration, the pivot dimension for R2D2 is Temperature at the considered grid cell.





- The Spatial configuration (S) aims at correcting the spatial correlations for each physical variable separately: each variable is corrected independently, and for each variable the $N_S$ vector of all values in the domain is corrected. Here, the pivot dimension for R2D2 is the considered physical variable at the center of the region.

- The Spatial-Intervar configuration (SI) intents to correct simultaneously the inter-variable and the spatial correlations of the simulations: the complete $5N_S$ vector of all variables in the domain is corrected at once. In this configuration, the pivot for R2D2 is the Temperature at the center of the region.

In addition to the historical reference data SAFRAN, for each of the three regions, there is thus a total of 8 climate datasets with the physical variables described in Sect. 2.1 (tas, pr, scfWind, rsds and hurs): IPSL (gridded to the SAFRAN 8 km × 8 km mesh) and 7 bias corrected datasets. Using those 9 datasets as input variables, ET0 and FWI are computed every day at all grid cells. Then, for each plant model and at all grid cells, the SWC is computed every day and FLO is determined for each year.

In this work, the summer season (92 days in June, July and August) has been selected for analysis because variations of ET0, SWC and FWI are expected to be amplified and differences between MBC configurations largest. In particular, the Provence region is characterized by high temperatures and low precipitations in Summer.

## 3.2 Statistical analysis

Let us denote $Z(\boldsymbol{s}, t)$, one of the output variable, computed at site $\boldsymbol{s} \in \mathcal{S}$ and Julian day $t$. There is a total of $n_S$ sites in $\mathcal{S}$ and $n_T$ Julian days considered for the analysis every year. Measurements (or computations) are repeated during $m$ years of a period whose climate is considered as being approximately constant. The $m$ years are thus assumed to be independent and identically distributed repetitions of the same spatio-temporal process. Several summary statistics, described below, are computed for visualization and hypothesis testing.

**Univariate statistics**

For a given spatio-temporal output $Z(\boldsymbol{s}, t)$, the mean and variances are computed at each site,

$$\hat{\mu}(\boldsymbol{s}) = (mn_T)^{-1} \sum_{i=1}^{m} \sum_{t_i=1}^{n_T} Z(\boldsymbol{s}, t_i); \qquad \hat{\sigma}^2(\boldsymbol{s}) = (mn_T)^{-1} \sum_{i=1}^{m} \sum_{t_i=1}^{n_T} \left( Z(\boldsymbol{s}, t_i - \hat{\mu}(\boldsymbol{s}) \right)^2, \tag{1}$$

where $t_i$ denotes the Julian day $t$ in year $i$. From these, biases and variances can be computed at each location $s$ according to

$$\text{Bias}(\boldsymbol{s}) = \hat{\mu}_M(\boldsymbol{s}) - \hat{\mu}_R(\boldsymbol{s}); \quad \text{VarRatio}(\boldsymbol{s}) = \hat{\sigma}_M^2(\boldsymbol{s}) / \hat{\sigma}_R^2(\boldsymbol{s}), \tag{2}$$

where the index $M$ refers to one of the models (with or without bias correction) and $R$ stands for 'Reference'. These local statistics can be represented as maps or summarized as boxplots, but in this case the spatial information is lost.

**Spatial covariance and Moran's I**

In order to assess the spatial structure in $Z(\boldsymbol{s}, t)$, the spatial auto-covariance at short distances is computed assuming second-order stationarity. It is known that given the size of the domains under consideration (from 16,000 km² to 22,000 km²) and





the complex topographic structures, in particular in Provence, one must expect that the mean and variance of $Z(\boldsymbol{s},t)$ vary in space. However, a locally stationary assumption is possible, at least in Brittany and Ile-de-France, at the 40 km scale, which

corresponds approximately to 5 SAFRAN grid meshes. For a given spatial lag $\boldsymbol{k} \in \{-5,-4,\ldots,4,5\} \times \{-5,-4,\ldots,4,5\}$ the empirical spatial covariance is

$$\hat{C}(\boldsymbol{k}) = (mn_T n_S(\boldsymbol{k}))^{-1} \sum_{i=1}^{m} \sum_{t_i=1}^{n_T} \sum_{\boldsymbol{s}\in\mathcal{S}_{\boldsymbol{k}}} \big(Z(\boldsymbol{s},t_i)-\hat{\mu}(\boldsymbol{s})\big)\big(Z(\boldsymbol{s}+\boldsymbol{k},t_i)-\hat{\mu}(\boldsymbol{s}+\boldsymbol{k})\big), \qquad (3)$$

where $\mathcal{S}_{\boldsymbol{k}}$ is the restriction of $\mathcal{S}$ with $n_S(\boldsymbol{k})$ elements such that both $\boldsymbol{s}$ and $\boldsymbol{s}+\boldsymbol{k}$ are in $\mathcal{S}$.

Moran's I (Moran, 1950) is a widely used measure of spatial auto-correlation at short distances. We use here a local, un-

normalized version of Moran's I given by

$$I = \frac{1}{mn_T \sum_{\boldsymbol{s}}\sum_{\boldsymbol{s'}} w_{\boldsymbol{s},\boldsymbol{s'}}} \sum_{i=1}^{m}\sum_{t=1}^{n_t}\sum_{\boldsymbol{s}}\sum_{\boldsymbol{s'}} w_{\boldsymbol{s},\boldsymbol{s'}} \big(Z(\boldsymbol{s},t_i)-\hat{\mu}(\boldsymbol{s})\big)\big(Z(\boldsymbol{s'},t_i)-\hat{\mu}(\boldsymbol{s'})\big), \qquad (4)$$

where $w(\boldsymbol{s},\boldsymbol{s'})$ is a binary indicator that characterizes the neighborhood structure with $w(\boldsymbol{s},\boldsymbol{s}) = 0$. The 'rook' (resp. 'queen') neighborhood corresponds to $||\boldsymbol{s}-\boldsymbol{s'}|| \leq 1$ (resp. to $||\boldsymbol{s}-\boldsymbol{s'}|| \leq \sqrt{2}$), where the distance is expressed in mesh units. The measure $I$ in Eq. (4) is local because local means $\hat{\mu}(\boldsymbol{s})$ are used, and contrarily to the usual Moran's I it is not normalized by the variance,

for an easier implementation of the hypothesis testing presented below. Using the symmetry of the covariance function, direct manipulations of Eq. (4) show that

$$I_{\text{rook}} = \frac{\hat{C}(0,1)+\hat{C}(1,0)}{2} \quad \text{and} \quad I_{\text{queen}} = \frac{\hat{C}(0,1)+\hat{C}(1,0)+\hat{C}(1,1)+\hat{C}(-1,1)}{4}, \qquad (5)$$

which shows that Moran's I is a summary of the short distance behavior of the spatial covariance function. The spatial covariance will be represented as a function of the distance $d = 8||\boldsymbol{k}||$ (in km), where $\boldsymbol{k}$ is the spatial lag vector between SAFRAN

grid meshes.

**Spatio-temporal correlation**

At larger scales, the spatio-temporal non-stationarity must be acknowledged. We thus decompose $Z(\boldsymbol{s},t)$ according to

$$Z(\boldsymbol{s},t) = \mu(\boldsymbol{s},t) + \sigma(\boldsymbol{s},t)\varepsilon(\boldsymbol{s},t), \qquad (6)$$

where the mean $\mu(\boldsymbol{s},t)$ and standard deviation $\sigma(\boldsymbol{s},t)$ vary in space and time. They are estimated with their empirical versions:


$$\hat{\mu}(\boldsymbol{s},t) = m^{-1}\sum_{i=1}^{m} Z(\boldsymbol{s},t_i); \qquad \hat{\sigma}^2(\boldsymbol{s},t) = m^{-1}\sum_{i=1}^{m}\big(Z(\boldsymbol{s},t_i)-\hat{\mu}(\boldsymbol{s},t)\big)^2. \qquad (7)$$

In Eq. (6), $\varepsilon(\boldsymbol{s},t)$ is a standardized residual. In all generality, the spatio-temporal correlation function $\text{Cor}\big(\varepsilon(\boldsymbol{s},t),\varepsilon(\boldsymbol{s'},t')\big) = \rho(\boldsymbol{s},\boldsymbol{s'},t,t')$ is any positive definite function of $(\boldsymbol{s},\boldsymbol{s'},t,t')$ (Chen et al., 2021). However, motivated by the absence of complex





space-time interactions in $\varepsilon(\boldsymbol{s},t)$, such as diffusion or transport, the spatio-temporal correlation function for $\varepsilon$ is assumed to be
space-time separable with

$$\mathrm{Cor}\big(\varepsilon(\boldsymbol{s},t),\varepsilon(\boldsymbol{s}',t')\big) = \rho(\boldsymbol{s},\boldsymbol{s}',t,t') = \rho_S(\boldsymbol{s},\boldsymbol{s}')\rho_T(t,t'). \tag{8}$$

The spatial and temporal correlations are estimated using temporal and spatial repetitions, respectively:

$$\hat{\rho}_S(\boldsymbol{s},\boldsymbol{s}') = (mn_T)^{-1}\sum_{i=1}^{n}\sum_{t=1}^{n_T}\mathrm{Cor}\big(\varepsilon(\boldsymbol{s},t),\varepsilon(\boldsymbol{s}',t)\big); \qquad \rho_T(t,t') = (mn_S)^{-1}\sum_{i=1}^{n}\sum_{\boldsymbol{s}\in\mathcal{S}}\mathrm{Cor}\big(\varepsilon(\boldsymbol{s},t),\varepsilon(\boldsymbol{s},t')\big). \tag{9}$$

We refer the reader to Chen et al. (2021) and references therein for an in-depth discussion on separability for spatio-temporal
correlation functions, its application and testing.

**Effective Sample Size**

When the sample values are spatially correlated, the actual number of data cannot be taken as such for computing the degrees
of freedom for hypothesis testing. One must instead assess the correlation between the values and derive an Effective Sample
Size (ESS), which quantifies the number of independent and identically distributed observations within the sample under
consideration. Let us consider a sample $\boldsymbol{Z}$ of size $n$ with common marginal expectation $\mu$ and variance $\sigma^2$ and with correlation
matrix $\boldsymbol{R}$. Then, under the assumption that $\boldsymbol{R}$ is invertible, Vallejos and Osorio (2014) define the ESS as $\mathrm{ESS} = \mathbf{1}_n^\top \boldsymbol{R}^{-1}\mathbf{1}_n$,
where $^\top$ is the transpose operator and $\mathbf{1}_n$ is a vector of 1s of length $n$. There is a enlightening interpretation to the ESS, in
relation to the estimation of $\mu$ when $\boldsymbol{R}$ is known. It can be shown that in this case the best (i.e. unbiased and with minimum
variance) estimator of $\mu$ is $\hat{\mu} = \mathbf{1}_n^\top \boldsymbol{R}^{-1}\boldsymbol{Z}$ and that its variance is $\mathrm{Var}(\hat{\mu}) = \sigma^2/\mathrm{ESS}$ (Chiles and Delfiner, 2012, Sect. 3.4).
ESS depends on $n$ and on the correlation structure of $\boldsymbol{Z}$ which, in a spatio-temporal context, depends on the space and time
coordinates of the samples and on the spatio-temporal correlation function. ESS decreases from $n$ to 1 as the correlation
strength decreases from no correlation (i.e. $\boldsymbol{R}$ is the identity matrix of size $n \times n$) to perfect correlation (i.e. $\boldsymbol{R}$ is the $n \times n$
matrix of 1s).

In the spatio-temporal context above, the correlation matrix to be considered is of size $n_S n_T \times n_S n_T$, which can be too
large for an easy inversion (for example, the summer season in Provence would yield to a $31,004 \times 31,004$ matrix). However,
under the separability assumption in Eq. (8), the computation of the SSE corresponding to one year of data simplifies to
$\mathrm{SSE}_1 = \mathrm{SSE}_S \times \mathrm{SSE}_T$, with $\mathrm{SSE}_S = \mathbf{1}_n^\top \boldsymbol{R}_S^{-1}\mathbf{1}_n$ and $\boldsymbol{R}_{S,ij} = \hat{\rho}_S(\boldsymbol{s}_i,\boldsymbol{s}_j)$, with $1 \le i,j \le n_S$, and with a similar expression for
$\mathrm{SSE}_T$. As an illustration, for the summer season in Provence, $n_S = 337$ and $n_T = 92$. Finally, the SSE of a given period (e.g.
the summer season) for $m$ independent years is simply $\mathrm{SSE}_m = m\mathrm{SSE}_1$.

**Hypothesis testing**

Two types of statistical tests are performed. The first type aims at testing the absence of bias or differences on global averages.
The basis for this is the two sample t-test with unequal variances (Snedecor and Cochran, 1989). The second family aims at
testing whether variances and Moran's I are equal or unequal. Fort this, the Fisher's F-tests of equality of variances, based





on the ratio of the variances (Snedecor and Cochran, 1989), is used. In all cases, an important parameter for these tests is the
'degrees of freedom', equal to $n-1$ when the $n$ samples are independent. Here, following the discussion in the paragraph
above, the degrees of freedom is set to $\mathrm{SSE}_m - 1$ to take into account the spatio-temporal auto-correlation.

## 4   Results

We present here a representative selection of the results allowing for interpretation and discussion. Recall that summer (JJA) has
been selected for analysis in this work. For size consideration, it is not possible to present all the results for all combinations
of indicators, plant models and regions, neither in the main text nor as usual Supplementary Material. A complete and un-
commented presentation of all results is accessible in the technical report Allard et al. (2023), which is freely accessible at
https://hal.inrae.fr/hal-04227826. This reference will serve as Supplementary Material and as a general rule, whenever not
shown results are reported or discussed, they can be found at this URL.

### 4.1   Phenological stage

Flowering (FLO) stage for the crop studied depends only on temperature. It is a cumulative indicator, the temporal dynamics is
thus important. It is worth recalling that for FLO stage, there is only one value per year, and that in some occasions (locations
and/or years) FLO stage is never attained, thus producing a NaN value in this case. As a consequence, the statistical tests have
less power and higher type II errors (not rejecting when the null hypothesis is not true) than for the other indicators. For FLO
stage, we shall therefore only comment the cases for which the hypotheses are rejected and not comment cases where the
p-value is higher than 0.1.

Summary statistics for biases, variance ratio and spatial covariances are shown in Fig. 2 for maize in Ile-de-France (a possible
dominant crop in the future, favored by increased temperatures) and for vine in Provence (currently, a major wine producer
region in France).

The IPSL climate model shows positive bias in Ile-de-France for all plant models, around 5 days for maize (Fig. 2(a) and up
to 8 days for vine (Allard et al., 2023, Fig. 4.1). This is partly due to the Paris region heatspot not properly taken into account
in GCMs (see Fig. 1middle-center).

In Provence, the situation is contrasted. On the one hand there are some pixels with very high negative bias due, mainly
in the mountainous areas where FLO stage is sometimes not attained due to high elevations, see e.g. Fig 4.11 in Allard et al.
(2023). On the other hand, there seems to be almost no bias for the majority of the pixels with low elevation. Overall, equality
of mean is nonetheless rejected.

For all bias correction methods, an almost complete reduction of bias can be observed in all regions. For maize in Ile-de-
France, the variance is slightly overestimated for all methods, except for I.dOTC and SI.dOTC for which it is underestimated
(Fig. 2(b). It is correctly reproduced for vine in Provence, except again for I.dOTC and SI.dOTC (2h). Spatial covariances are
flat in Ile-de-France (2c) because values are highly correlated in space. There is more spatial structure in Provence (2i), and
it is correctly reproduced for CDF-t and I.R2D2. The spatial structure is broken for IPSL, and otherwise well reproduced (up

| | | IPSL | CDF-t/I.R2D2 | I.dOTC | S.dOTC | S.R2D2/SI.R2D2 | SI.dOTC |
|---|---|---|---|---|---|---|---|
| | | | p-values for "equality-of-means" tests | | | | |
| Britt. | wheat | 0.052 | **0.856** | **0.676** | **0.976** | **0.872** | **0.371** |
| | maize | **0.626** | **0.788** | **0.909** | **0.981** | **0.828** | **0.578** |
| | vine | **1.000** | **0.807** | **0.906** | **0.931** | **0.850** | **0.465** |
| IdF | wheat | 0.016 | **0.820** | **0.953** | **0.820** | **0.837** | **0.978** |
| | maize | 0.009 | **0.866** | **0.765** | **1.000** | **0.897** | **0.583** |
| | vine | 0.000 | **0.759** | **0.993** | **0.891** | **0.785** | **0.988** |
| Prov. | wheat | 0.000 | **0.831** | 0.001 | **0.243** | **0.894** | 0.000 |
| | maize | 0.000 | **0.106** | 0.003 | 0.018 | **0.114** | 0.004 |
| | vine | 0.000 | 0.000 | 0.000 | 0.001 | 0.002 | 0.000 |
| | | | p-values for "equality-of-variances" tests | | | | |
| Britt. | wheat | **0.439** | **0.973** | 0.006 | **0.899** | **0.869** | 0.000 |
| | maize | **0.699** | **0.160** | 0.056 | **0.214** | **0.334** | 0.000 |
| | vine | **1.000** | 0.062 | **0.159** | **0.110** | **0.195** | 0.000 |
| IdF | wheat | **0.887** | **0.961** | 0.096 | **0.975** | **0.951** | 0.000 |
| | maize | **0.594** | **0.452** | **0.195** | **0.520** | **0.548** | 0.000 |
| | vine | **1.000** | **0.341** | **0.210** | **0.405** | **0.446** | 0.000 |
| Prov. | wheat | 0.018 | 0.002 | 0.000 | 0.000 | 0.000 | 0.000 |
| | maize | 0.000 | **0.368** | 0.000 | **0.117** | **0.397** | 0.000 |
| | vine | **1.000** | 0.058 | 0.000 | 0.014 | 0.033 | 0.000 |
| | | | p-values for "equality-of-Moran's I" tests | | | | |
| Britt. | wheat | **0.448** | **0.964** | 0.003 | **.0.901** | **0.882** | 0.000 |
| | maize | **0.688** | **0.147** | 0.017 | **0.199** | **0.299** | 0.000 |
| | vine | **1.000** | 0.051 | 0.099 | 0.093 | **0.163** | 0.000 |
| IdF | wheat | **0.865** | **0.947** | 0.072 | **0.990** | **0.968** | 0.000 |
| | maize | **0.556** | **0.432** | **0.105** | **0.502** | **0.526** | 0.000 |
| | vine | **1.000** | **0.333** | **0.115** | **0.395** | **0.434** | 0.000 |
| Prov. | wheat | **0.101** | 0.012 | 0.000 | 0.000 | 0.000 | 0.000 |
| | maize | 0.006 | **0.655** | 0.000 | **0.209** | **0.577** | 0.000 |
| | vine | **1.000** | 0.058 | 0.000 | 0.014 | 0.033 | 0.000 |

**Table 2.** Statistical analysis for FLO in the past: p-values for the Welsh t-test of absence of bias on the average (first block); Fisher F-test of equality of variance (second block) and its adaptation to testing the equality of Moran's I (third block). Non rejection at the confidence level 0.90 is indicated in bold font. Since FLO is univariate, CDF-t and I.R2D2 with Temperature as pivot variable are equivalent.

to the variance multiplicative effect) for all bias correction methods. Overall, equality of variance and equality of Moran's I is always rejected with SI.dOTC and quite often rejected with I.dOTC (Table 2).



**Figure 2.** Results for FLO (unit is day) for maize in Ile-de-France (from a) to f)) and grapevine in Provence (from g) to l)). First and third rows (from a) to c) and g) to i)) correspond to past period with: a) and g) boxplots of differences to SAFRAN; b) and h) boxplots of variance ratios to SAFRAN; c) and i) spatial covariance (colored points, covariance as a function of distance) and Moran's I (points on the vertical dashed line). Second and fourth rows (from d) to f) and j) to l) correspond to future period with: d) and j) boxplots of differences between future and past; e) and k) boxplots of variance ratios between future and past; f) and l) spatial covariance and Moran's I ratios between future and past.





Since FLO stage considers temperature in a cumulative way, the temporal dynamics of the single variable temperature is key for the interpretation of theses results. Whenever the results are accurate (in terms of bias or (co-)variance ratio) one can

consider that the marginals are well corrected with appropriate temporal dynamics from the climate model. CDF-t (as well as I.R2D2 which is equivalent in this univariate setting) does not change the dynamics of the model in terms of ranks. Since CDF-t provides unbiased results for FLO stage with an accurate variance ratio in both regions, one can conclude that the dynamics of the IPSL model is in accordance with the reanalysis for Temperature and hence for FLO stage and that CDF-t provides the correct correction of the bias.

By construction, dOTC will try to correct the marginals and the copulas (intervariable and/or spatial) while not changing too much the dynamics of the model (Robin et al., 2019). Therefore, in I.dOTC, S.dOTC and SI.dOTC configurations, the temporal dynamics of Temperature will be modified when correcting for the intervariable and/or spatial copula. As a general rule, it was shown in François et al. (2020, Fig. 1, 5 and S5) that the more the variables are to be corrected using dOTC, the more both the marginals and the temporal dynamics are modified. This result was also partly observed here: as the number of variables to

correct increases from I.dOTC to SI.dOTC, a slight but significant increasing bias is observed for maize in Ile-de-France (Fig. 2(a)). This also applies for the variance of the I.dOTC and SI.dOTC outputs that is strongly underestimated in both regions (Fig. 2(b)). Nevertheless, quite interestingly, the increasing deterioration with respect to the dimensional setting is not observed for S.dOTC. In this setting, each variable is spatially corrected in turn. Given the high spatial autocorrelation of temperature, dimensions are then somehow redundant, which potentially reduces the number of "effective dimensions" to adjust and so the

complexity of the correction to provide. Consequently, the impact of dOTC on the temporal rank correlation is less pronounced, and better results are obtained with S.dOTC. These results illustrate that when using dOTC, one must take care to correct only the variables that are involved in the process under study.

Overall, results for all R2D2 methods are good, both in terms of bias and variance ratio (Fig. 2(a,b,g,h)). In comparison to dOTC, R2D2 preserves the temporal rank structure of the pivot variable, which here is temperature: at each pixel for I.R2D2

and at the center of the domain for S.R2D2 and SI.R2D2. In conjunction with the fact that the spatial rank autocorrelation for Temperature is very high (not shown), the temporal rank structure is either exactly (I.R2D2) or partially well (S.R2D2 and SI.R2D2) preserved. Hence, structures in the spatial covariances are well reproduced (Fig. 2(c,i)), but an overall overestimation in relation to the variance ratio is observed (Fig. 2(b,h)). Notice that if a different pivot variable was chosen, the temporal dynamics would have been stirred, and therefore less comparable to the reanalysis, see François et al. (2020, Fig. S5) for

an illustration of the modification of the temporal properties of Temperature when Precipitation is the pivot variable. Similar results were obtained for the other region × plant models configurations (Allard et al., 2023, Chap. 4 and 5).

When looking at differences and variance ratios in the Future, one can observe an important advance of FLO for all plant models, regions and methods, from about 9 days for maize in Ile-de-France (Fig. 2(d) to 2 weeks for vine in Provence (Fig. 2(j)). Overall, this difference is similar for all bias correction methods, which is reminiscent of the very similar corrections for

all bias correction methods in the Past. In relation with the strong underestimation of the variance for I.dOTC and SI.dOTC, the variance ratio between future and past are higher for these methods than for the others (Fig. 2(e,k)). When looking at the maps





for Future (Allard et al., 2023, Fig. 5.12 to 5.20), one can see that the Future/Past variance ratios show a strong pixel pattern for I.dOTC and SI.dOTC.

## 4.2 Evapotranspiration ET0

ET0 is a climatic indicator computed every day using all variables of that day, except precipitation. It is thus multivariate but rather insensitive to the temporal dynamics. Summary statistics computed in Provence are shown in Fig. 3 and p-values in all regions are reported in Table 3. IPSL is strongly biased with a negative error close to 50 mm, its variance is strongly underestimated as well as its Moran's I and its spatial covariance (3, respectively panels a,b,c).

The bias, the variance and the spatial covariance are in general well corrected for all methods. Differences between methods
are nonetheless visible on the variance ratios and the spatial covariances. The variance is overestimated with CDF-t and S.R2D2 (in Provence, p-values of Fisher's F-tests are equal to 0.000 and 0.011, respectively) and the spatial covariance structure is broken with I.R2D2, as evidenced by Morans'I values being divided by a factor ∼3 with respect to SAFRAN (Fig. 3(c)). Similar results were obtained in the other two regions Allard et al. (2023, chap. 2) in terms of bias, variance and spatial covariance structure, but to a lesser extent: the bias is not as large, equality of variance is not rejected and Morans'I as well as
the spatial covariance are strongly underestimated.

Since CDF-t provides an unbiased correction with a variance ratio close to one, even in Provence, one can conclude that the inter-variable copula of the IPSL model is rather similar to that of the reanalysis. However, ET0 being a multivariate indicator, multivariate correction methods are expected to lead to improved variance and Moran's I than CDF-t. It is indeed the case in Provence where the p-value for the equality of variance is equal to 0 with CDF-t and larger than 0.1 for all methods, except for
S.R2D2. S.R2D2 reshuffles the ranks of the IPSL model to match the rank pattern of the reanalysis (the pivot is at the center of the domain) independently for each variable. As a result, the variance and the spatial covariance are slightly increased, in particular in Provence, which is highly non-stationary.

As already pointed out, in Provence, Moran's I is reduced by a factor of three when using I.R2D2 (Fig. 3(c)). The reason for I.R2D2's poor spatial behavior is that its rank shuffling is performed separately at each pixel, thereby breaking the spatial
correlation present in the IPSL model for all variables other than the pivot. In contrast, the spatial structure is well reproduced with SI.R2D2 since in this case the inter-variable and the spatial correlations are taken into account in the correction. All dOTC bias corrections perform well in terms of bias, variance and spatial covariance, even for I.dOTC. This is due to the fact that I.dOTC partially preserves the spatial dependencies of the IPSL model (both Intervariable and spatial) as shown in Robin et al. (2019), which is also evidenced by the fact that the spatial structure is well reproduced with CDF-t.

When looking at future ET0 values according to the IPSL model, one can notice that there is almost no difference between ET0 in the future and ET0 in the past (Fig. 3(d)). In contrast, for all bias correction methods there is a daily difference from 7 to 11 mm evapotranspiration in Provence between Future and Past. This result was also observed in the other two regions (Allard et al., 2023, Fig. 3.1). As a first observation, one notices that a rather small positive difference (around 1 mm) of ET0 between future and past with the IPSL model is amplified to a 7 to 10 mm difference after corrections, depending on the bias
correction method. As a consequence, as ET0 is expected to increase in a warmer climate in temperate regions such as France





(Lemaitre-Basset et al., 2022), projected ET0 computed from IPSL is thus likely to be even more biased than in the past. The variance ratio between future and past ranges between 0.8 and 0.9 for all bias correction methods, and it is as low as 0.75 according to IPSL (Fig. 3(e)). However, the variance underestimation is less pronounced in the other two regions, which are located in the Northern part of France with less severe evapotranspiration during the summer (Allard et al., 2023, Fig. 3.1). As

a last remark, maps of variance ratios between future and past show important spatial patterns (Allard et al., 2023, Fig. 3.6 to 3.8). Also, these maps are less regular for dOTC methods than for R2D2 methods.

|  | IPSL | CDF-t | I.dOTC | I.R2D2 | S.dOTC | S.R2D2 | SI.dOTC | SI.R2D2 |
|---|---|---|---|---|---|---|---|---|
| | | | p-values for "equality-of-means" tests | | | | | |
| Britt. | 0.000 | **0.396** | **0.826** | **0.967** | **0.817** | **0.773** | **0.828** | **0.966** |
| IdF | 0.000 | **0.468** | **0.820** | **0.958** | **0.923** | **0.931** | **0.822** | **0.951** |
| Prov. | 0.000 | **0.812** | **0.664** | **0.804** | **0.808** | **0.932** | **0.609** | **0.749** |
| | | | p-values for "equality-of-variances" tests | | | | | |
| Britt. | 0.000 | **0.101** | **0.883** | **0.803** | **0.729** | **0.877** | **0.885** | **0.795** |
| IdF | 0.000 | **0.212** | **0.891** | **0.779** | **0.940** | **0.625** | **0.892** | **0.746** |
| Prov. | 0.000 | 0.000 | **0.860** | **0.439** | **0.163** | 0.011 | **0.834** | **0.324** |
| | | | p-values for "equality-of-Moran's I" tests | | | | | |
| Britt. | 0.000 | 0.004 | **0.962** | 0.000 | **0.673** | **0.771** | **0.885** | **0.795** |
| IdF | 0.000 | 0.024 | **0.706** | 0.000 | **0.928** | **0.631** | **0.893** | **0.744** |
| Prov. | 0.000 | 0.000 | **0.604** | 0.000 | **0.120** | 0.007 | **0.837** | **0.330** |

**Table 3.** Statistical analysis for summer ET0 in the past: p-values for the Welsh t-test of absence of bias on the average (first block); Fisher F-test of equality of variance (second block) and its adaptation to testing the equality of Moran's I (third block). Non rejection at the confidence level 0.90 is indicated in bold font.

## 4.3 Soil Water Content

The Soil Water Content (SWC) is computed every day using the SWC of the previous day and all physical variables of the given day. It is thus multivariate and sensitive to the temporal dynamics. The usual summary statistics are shown for maize

in Ile-de-France in Fig. 4. IPSL leads to strongly biased values of SWC and overestimated variances and Moran's I for all combinations of plant models and regions (Fig. 4(a)), only shown for maize in Ile-de-France.

Table 4 reports the p-values for equality of means, variances and Moran's I, for all models and regions. In general, the bias is more easily corrected than the variance, which in turn is more easily corrected than Moran's I. According to the p-values, Provence seems to be a more difficult region to correct than Brittany and Ile-de-France for all methods, presumably because

it is spatially nonstationary. Most bias correction methods do a fair job at correcting the bias, except I.R2D2 and SI.R2D2 in Brittany and Ile-de-France, as well as CDF-t in Provence. Notice however that even though equality of means in Provence is rejected at the level 0.90 for maize and vine, it is not rejected at the more conservative level 0.95.

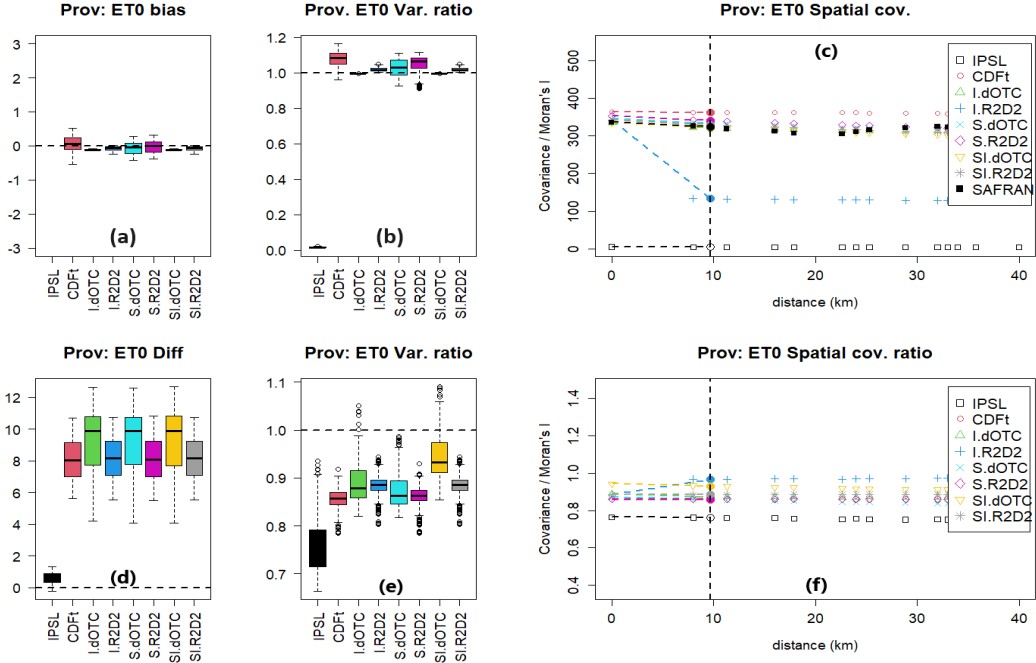

**Figure 3.** Results for summer ET0 (unit is mm) in Provence. First row corresponds to past period with: a) boxplot of differences to SAFRAN (the bias for IPSL, varying between $-45$ and $-50$, lies out of the range of the plot); b) boxplot of variance ratios to SAFRAN; c) spatial covariance (colored points, covariance as a function of distance) and Moran's I (points on the vertical dashed line). Second row corresponds to future period with: d) boxplot of differences between future and past; e) boxplot of variance ratios between future and past; f) spatial covariance and Moran's I ratios between future and past.

Overall, SI.dOTC and I.dOTC methods perform better than CDF-t and S.dOTC, because the highly multivariate aspect of SWC benefits from the multivariate correction without too much perturbation of the temporal dynamics of the model. In contrast, I.R2D2 and SI.R2D2 induce a strong perturbation of the temporal dynamics of all variables except the pivot variable (here Temperature) impairing a proper reproduction of the variance and the Moran's I of SWC, which is strongly underestimated for I.R2D2 (see also Fig. 4(b)). Despite many p-values lower than 0.1, the boxplots of the variance ratio for S.R2D2 show however a fairly decent reproduction of the variance and the Moran's I. Quite interestingly, S.R2D2 leads to a more accurate correction than I.R2D2 and SI.R2D2 even though it does not adjust the inter-variable properties. This result might seem counter-intuitive and it deserves some explanation. When analyzing the outputs of a time-integrative and multivariate process such as SWC, the relative importance of each dimension (time vs. intervariable) plays an important role. It is well known that I.R2D2 and, to a lesser extent, SI.R2D2, alter the temporal dynamics (François et al., 2020). In contrast, since S.R2D2 is applied to each variable in turn, the temporal dynamics is not too much altered when the spatial correlation is important, which is the case for the CMIP6-IPSL model in Brittany and Ile-de-France. Our interpretation of this result is





that the time dynamics plays a more important role for SWC than the multivariate aspect, when it comes to multivariate bias corrections.

The CMIP6-IPSL model indicates lower values of SWC and decreased variances in the future, see Fig. 4(d,e) for maize in Ile-de-France, and Allard et al. (2023, chap. 7) for other combinations of plant models and all regions. Generally speaking, bias correction methods tend to lessen these differences, in particular for I.dOTC and SI.dOTC. Recall that these methods where

among the most accurate bias correction methods in the past period. Maps of SWC variance ratio between future and past (Fig. 5) show that for I.R2D2 the variance of SWC is highly variable at short distances, in accordance with the very low value of Moran's I metrics in Fig. 4(c). The variance ratio varies much more smoothly when correcting with SI.dOTC.

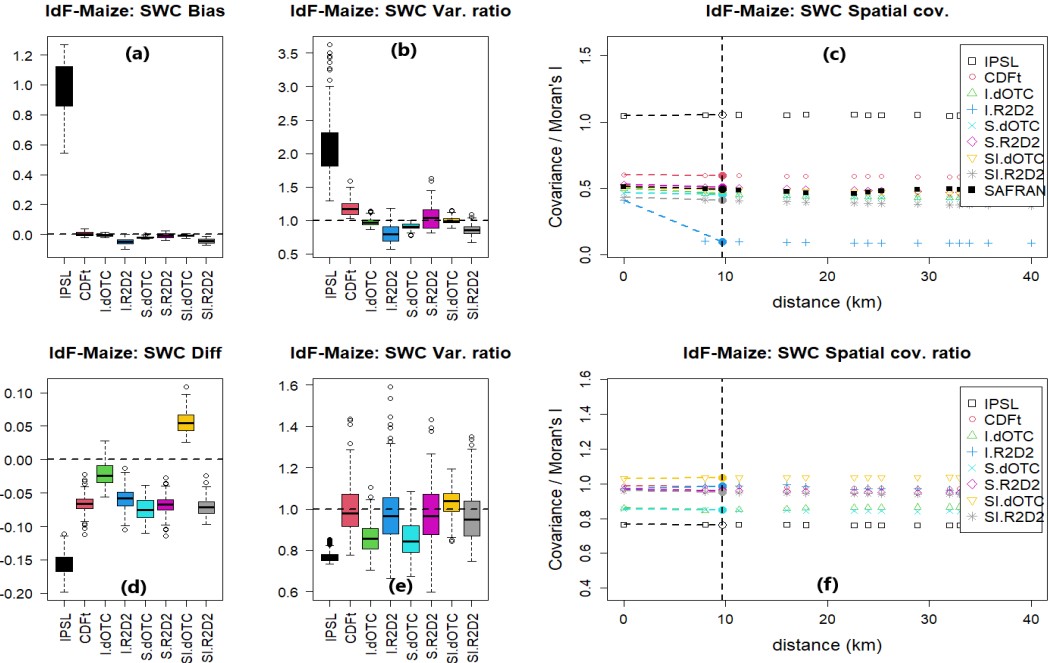

**Figure 4.** Results for summer SWC (unit is %) in Ile-de-France and maize. First row corresponds to past period with: a) boxplot of differences to SAFRAN; b) boxplot of variance ratios to SAFRAN; c) spatial covariance (colored points, covariance as a function of distance) and Moran's I (points on the vertical dashed line). Second row corresponds to future period with: d) boxplot of differences between future and past; e) boxplot of variance ratios between future and past; f) spatial covariance and Moran's I ratios between future and past.





|  |  | IPSL | CDF-t | I.dOTC | I.R2D2 | S.dOTC | S.R2D2 | SI.dOTC | SI.R2D2 |
|---|---|---|---|---|---|---|---|---|---|
| | | \multicolumn{8}{c}{p-values for "equality-of-means"} | | | | | | | |
| Britt. | wheat | 0.000 | **0.570** | **0.798** | 0.005 | **0.467** | **0.137** | **0.151** | 0.002 |
| | maize | 0.000 | 0.053 | **0.370** | 0.000 | **0.446** | **0.661** | **0.766** | 0.000 |
| | vine | 0.000 | **0.312** | **0.634** | 0.000 | **0.983** | **0.335** | **0.570** | 0.000 |
| IdF | wheat | 0.000 | **0.773** | **0.770** | **0.131** | **0.339** | **0.567** | **0.441** | **0.133** |
| | maize | 0.000 | **0.884** | **0.778** | 0.002 | **0.290** | **0.634** | **0.620** | 0.006 |
| | vine | 0.000 | **0.407** | **0.724** | 0.002 | **0.503** | **0.577** | **0.749** | 0.000 |
| Prov. | wheat | 0.000 | 0.024 | **0.342** | **0.598** | **0.387** | **0.343** | **0.982** | **0.409** |
| | maize | 0.000 | 0.091 | **0.688** | **0.550** | **0.728** | **0.841** | **0.685** | 0.000 |
| | vine | 0.000 | 0.071 | **0.726** | **0.269** | **0.944** | **0.988** | **0.492** | **0.176** |
| | | \multicolumn{8}{c}{p-values for "equality-of-variances"} | | | | | | | |
| Britt. | wheat | 0.000 | 0.000 | **0.304** | 0.000 | 0.024 | **0.596** | 0.000 | 0.000 |
| | maize | 0.000 | 0.000 | 0.000 | 0.000 | 0.000 | 0.027 | **0.276** | 0.000 |
| | vine | 0.000 | 0.000 | **0.955** | 0.000 | **0.130** | 0.053 | 0.000 | 0.000 |
| IdF | wheat | 0.000 | **0.392** | 0.016 | 0.000 | 0.001 | 0.016 | 0.051 | 0.000 |
| | maize | 0.000 | 0.000 | **0.520** | 0.000 | 0.013 | **0.352** | **0.825** | 0.000 |
| | vine | 0.000 | 0.000 | **0.214** | 0.000 | 0.007 | **0.461** | 0.020 | 0.000 |
| Prov. | wheat | 0.000 | 0.000 | 0.000 | 0.050 | 0.000 | 0.000 | **0.526** | 0.000 |
| | maize | 0.000 | 0.000 | 0.000 | 0.003 | 0.000 | 0.000 | **0.880** | 0.000 |
| | vine | 0.000 | 0.000 | 0.012 | 0.000 | 0.000 | 0.007 | 0.007 | 0.000 |
| | | \multicolumn{8}{c}{p-values for "equality-of-Moran's I"} | | | | | | | |
| Britt. | wheat | 0.000 | 0.000 | **0.737** | 0.000 | 0.015 | **0.828** | 0.000 | 0.000 |
| | maize | 0.000 | 0.000 | 0.015 | 0.000 | 0.000 | 0.014 | **0.282** | 0.000 |
| | vine | 0.000 | 0.000 | **0.214** | 0.000 | **0.116** | 0.090 | 0.000 | 0.000 |
| IdF | wheat | 0.000 | **0.143** | 0.001 | 0.000 | 0.001 | 0.011 | 0.051 | 0.007 |
| | maize | 0.000 | 0.000 | 0.036 | 0.000 | 0.014 | **0.403** | **0.863** | 0.000 |
| | vine | 0.000 | 0.000 | 0.007 | 0.000 | 0.007 | **0.331** | 0.017 | 0.000 |
| Prov. | wheat | 0.000 | 0.000 | 0.000 | 0.000 | 0.000 | 0.000 | **0.203** | 0.000 |
| | maize | 0.000 | 0.000 | 0.000 | 0.003 | 0.000 | 0.000 | **0.570** | 0.000 |
| | vine | 0.000 | 0.000 | 0.012 | 0.000 | 0.000 | 0.006 | 0.008 | 0.000 |

**Table 4.** Statistical analysis for summer SWC in the past. p-values for the Welsh t-test of absence of bias on the average (first block); the Fisher F-test of equality of variance (second) block and its adaptation to testing the equality of Moran's I (third block). Non rejection at the confidence level 0.90 is indicated in bold font.

**Figure 5.** For each bias correction method and for maize: map of summer SWC variance ratio between future and past in Ile-de-France (no unit). From top to bottom and from left to right: IPSL (no correction), CDF-t, Intervar-dOTC, Intervar-R2D2, spatial-dOTC, spatial R2D2, spatial-intervar-dOTC, spatial-intervar-R2D2.





## 4.4 Fire Weather Index

As for SWC, FWI is computed on a daily basis using FWI of the previous day and all physical variables of the current day. It is thus also multivariate and sensitive to the temporal dynamics. Summary statistics are shown in Fig. 6 for Ile-de-France and Provence. The latter experiences high wildfires activity every year. It is not yet the case for the former, but recent projections show that it will be the case in the future (Fargeon et al., 2020; Galizia et al., 2023; Pimont et al., 2023). Table 5 reports the p-values for equality of means, variances and Moran's I for all regions.

There is a positive bias and an overestimation of the variance in Ile-de-France (Fig. 6(a,b,c)), a region with currently low values for FWI and low variance. There is negative bias and an underestimation of the variance in Provence (Fig. 6(g,h,i)), a region with high values for FWI and higher variance. Brittany shows results very similar to those of Ile-de-France (Allard et al., 2023, Chap. 8). Overall, the bias is well corrected for all methods (Fig. 6(a,g)), except I.R2D2 and SI.R2D2 in all regions (all p-values for all tests are equal to 0.0 for these two methods) and S.R2D2 in Provence (the p-value is equal to 0.008). The fact that CDF-t leads to a fair correction of the mean, in particular in Provence, is the indication that, once the marginals being corrected, the interdependence between the variables and the temporal dynamics is well represented in the IPSL model.

As it was also the case for SWC, I.dOTC and SI.dOTC methods perform better than CDF-t and S.dOTC (Fig. 6(a,g) and Table 5), because the highly multivariate aspect of FWI benefits from the multivariate correction without perturbing too much the temporal dynamics of the model. S.dOTC, which corrects each variable independently, performs slightly less well in Provence.

In contrast, I.R2D2 and SI.R2D2 induce a strong perturbation of the temporal dynamics impairing a proper correction of both the bias and the variance. In Provence, where FWI can reach high values and is spatially heterogeneous, the bias and the variance is less corrected with S.R2D2 than with the inter-variable versions of R2D2 (Fig. 6(g)), because the multivariate copula is perturbed by the independent corrections made by S.R2D2 on each variables. However, in Ile-de-France where FWI is low and spatially homogeneous, it is the opposite: S.R2D2 leads to more accurate corrections than I.R2D2 and SI.R2D2 (Fig. 6(a)). As a result, bias corrections on FWI thus behave like bias corrections on SWC in Ile-de-France due to spatial homogeneity. It is not the case in Provence, due to the highly non-stationary behavior of all variables involved. As already seen on all other indicators, the spatial structure is broken in both regions with I.R2D2 (Fig. 6(c,i)) because the rank shuffling is performed at each location independently.

FWI definitely increases in the Future in all regions, for the IPSL model, and for all bias correction methods (Fig. 6(d,j)). The same observation can be made for the variance. Focusing on the Provence region, the average FWI increases by approximately 1 to 4 units, depending on the bias correction method (see also the maps in Fig. 7), and the variance is roughly multiplied by 2. As a point of comparison, the relationship between the severity of wildfires and FWI change from linear to exponential when FWI is larger than 20 (Pimont et al., 2021). Figure 8 shows that the variance increases more in the North-Eastern parts of the region, which are regions of high mountains. As already observed on other outputs, the map of the variance ratios between future and past is much smoother with CDF-t and SI.dOTC than with other methods.





|  | IPSL | CDF-t | I.dOTC | I.R2D2 | S.dOTC | S.R2D2 | SI.dOTC | SI.R2D2 |
|---|---|---|---|---|---|---|---|---|
| | | | p-values for "equality-of-means" | | | | | |
| Britt. | 0.000 | 0.000 | **0.494** | 0.000 | **0.144** | **0.830** | **0.178** | 0.000 |
| IdF | 0.000 | 0.058 | **0.573** | 0.000 | **0.232** | **0.605** | **0.930** | 0.000 |
| Prov. | 0.000 | **0.303** | **0.482** | 0.040 | **0.124** | 0.008 | **0.265** | 0.011 |
| | | | p-values for "equality-of-variances" | | | | | |
| Britt. | 0.000 | 0.000 | **0.976** | 0.000 | 0.071 | 0.043 | 0.021 | 0.000 |
| IdF | 0.000 | 0.000 | **0.298** | 0.000 | 0.004 | **0.299** | **0.915** | 0.000 |
| Prov. | 0.000 | 0.000 | 0.021 | 0.002 | 0.000 | 0.000 | 0.004 | 0.000 |
| | | | p-values for "equality-of-Moran's I" | | | | | |
| Britt. | 0.000 | 0.000 | 0.035 | 0.000 | 0.084 | 0.059 | 0.028 | 0.000 |
| IdF | 0.000 | 0.000 | **0.354** | 0.000 | 0.001 | **0.466** | **0.803** | 0.000 |
| Prov. | 0.000 | 0.001 | 0.000 | 0.000 | 0.000 | 0.000 | 0.004 | 0.000 |

**Table 5.** Statistical analysis for summer FWI in the past: p-values for the Welsh t-test of absence of bias on the average (first block); Fisher F-test of equality of variance between future and past (second block) and its adaptation to testing the equality of Moran's I (third block). Non rejection at the confidence level 0.90 is indicated in bold font.





**Figure 6.** Results for summer FWI (no unit) in Ile-de-France (from a) to f)) and in Provence (from g) to l)). First and third rows (from a) to c) and g) to i)) correspond to past period with: a) and g) boxplots of differences to SAFRAN; b) and h) boxplots of variance ratios to SAFRAN; c) and f) spatial covariance (colored points, covariance as a function of distance) and Moran's I (points on the vertical dashed line). Second and fourth rows (from d) to f) and j) to l) correspond to future period with: d) and j) boxplots of differences between future and past; e) and k) boxplots of variance ratios between future and past; f) and l) spatial covariance and Moran's I ratios between future and past. Notice that in panel g), the boxplot extends to $-4.2$ with a lowesy value equal to $-6.4$.



**Figure 7.** For each bias correction method: map of the summer FWI difference between future and past (no unit). From top to bottom and from left to right: IPSL (no correction), CDF-t, Intervar-dOTC, Intervar-R2D2, spatial-dOTC, spatial R2D2, spatial-intervar-dOTC, spatial-intervar-R2D2.



**Figure 8.** For each bias correction method: map of the summer FWI variance ratio between future and past (no unit). From top to bottom and from left to right: IPSL (no correction), CDF-t, Intervar-dOTC, Intervar-R2D2, spatial-dOTC, spatial R2D2, spatial-intervar-dOTC, spatial-intervar-R2D2.



## 5   Conclusion and Discussion

In this work, we have tested two MBC methods, namely R2D2 (Vrac, 2018) and dOTC (Robin et al., 2019), to adjust the inter-variable and/or spatial dependence structures of 5 physical variables that are input variables for process models having different characteristics in terms of inter-variable dependencies and time integration. For each MBC method, three different configurations were considered (intervariable, spatial and a spatial-intervariable) to disentangle the relative effect of the various dependence structures. A univariate bias correction method (CDF-t, Michelangeli et al. (2009)) was also included in the study to assess the potential added value of MBC methods.

We have applied these 7 correction methods to IPSL-CM6A-LR model simulations with respect to the SAFRAN reanalysis data over three contrasted regions in France, four processes (FLO, ET0, SWC and FWI) and three plant models for FLO and SWC. This work is thus an extension to plant and forest processes of the study in François et al. (2020), which focused on climate indicators. To assess the corrections, several statistics have been computed to quantify the differences in means, variances, spatial covariances and Moran's I, on all impact indices. Significance of differences (or ratios) have been formally assessed using statistical tests. A key element for hypothesis testing on highly correlated domains is the notion of Effective Sample Size, which quantifies the number of equivalent independent observations within the domain of observation. These developments constitute a first methodological result, which can be of great use in further studies.

Generally speaking, the averages of the model outputs are better corrected than their variances, which, in turn, are easier to correct than the associated Morans'I and spatial covariance. Our study thus highlights that when assessing a bias correction method, one should not use average differences as only metric.

Our results show that there is no single best MBC method. Depending on the process under study and on the metrics considered, there can be a method performing better (or worst) than the other ones – or not. However, as shown in Sect. 4, it is possible to draw partial conclusions and to provide useful recommendations regarding the use of MBCs when multivariate and time dependent processes are involved. These conclusions and recommendations, summarized in Tables 6 and 7 respectively, are now discussed.

As a first finding, it must be highlighted that CDF-t does a fair job in many situations. It suffers from no major defects, but it can be improved in some situations: all MBC methods perform better than CDF-t when the process is multivariate with no time integration (such as ET0) and SI.dOTC slightly outperforms CDF-t for multivariate processes with time integration (such as SWC and FWI). This result can be interpreted as evidence that the temporal dynamics in IPSL-CM6A-LR model is mostly well represented. The inter-variable properties are also fairly well represented (otherwise CDF-t would perform poorly for multivariate processes) but multivariate corrections are nonetheless useful, in particular over spatially heterogeneous regions such as Provence.

When the process is univariate, such as FLO, multivariate bias correction methods are not recommended. With R2D2 methods, if the pivot variable is the physical variable under consideration (which was the case here), there are un-necessary computations but otherwise no harm. If the pivot variable is any other variable, the variable under consideration will be reshuffled, thereby inducing potentially large errors in the temporal dynamics of the process outputs. With dOTC methods, the multivari-





ate adjustment entails a less relevant modification of the marginal of the variable under consideration, and hence less accurate
corrections, as could be seen on FLO.

When the process is highly multivariate, such as ET0, SWC and FWI, multivariate corrections are expected to reduce the
bias. However, care must be taken as to which method should be used for a proper reproduction of the variance and Moran's
I. Generally speaking, R2D2 methods have stronger impact on the overall dependence structure (multivariate copula and/or
spatial covariance) than dOTC methods. Indeed, while dOTC "bends" the whole copula from the model to the reference, R2D2
does not change the pivot variable, but makes more dramatic changes to the other variables. As a consequence:

- When the temporal dynamics is not important (as is the case for ET0), most methods improve on CDF-t and perform
  well, at the exception of purely spatial configurations (S) when proper intervariable properties are essential to the process
  under study.

- When the process is multivariate and the temporal dynamics is important (for example for cumulative processes such
as SWC and FWI), I.R2D2 and SI.R2D2 should be avoided altogether because the Intervariable correction entails large
  perturbations in the temporal dynamics of the ranks for all variables but the pivot. When the process is cumulative,
  the bias and the variance are not properly corrected with these methods. Note however that this effect has been partly
  corrected in a recent extension of the R2D2 method (Vrac and Thao, 2020, R2D2 v2.0), for which the analog search is
  constrained to improve temporal properties.

- For similar reasons, I.R2D2 and SI.R2D2 should be avoided when the spatial structure is important, as is for example
  the case for SWC and FWI when the focus is on water and wildfires management at the regional scale. Indeed, it was
  found that I.R2D2 and SI.R2D2 break down the spatial structure of the indicator. This effect is clearly visible in Allard
  et al. (2023, Fig. 7.12 to 7.20 and 9.6 to 9.9).

When all dimensions are important (multivariate, time cumulative and spatial), there are only few options. As already
mentioned, CDF-t does a fair job because, contrarily to the marginals, the temporal dynamics and the intervariable copula
in the IPSL-CM6A-LR model are preserved, but at a coarser spatial resolution. Bias-correcting with univariate correction
methods, such as CDF-t, is a no-risk / no-gain option. However, SI.dOTC brings improvements in most cases and no significant
improvement in some rare cases. We did not find any multivariate cases where SI.dOTC performs less well than CDF-t.

Despite differences in the experiments (different climate variables, processes and BC methods), our results concur in gen-
eral with the main findings reported in Galmarini et al. (2024) and in Ahn et al. (2023), but we also found some important
differences:

- As already underlined above, compared to univariate bias correction methods, together with these two studies we found
  that multivariate methods improve the adjustment of model outputs to the reference in many cases, but there are excep-
  tions depending on the method considered, its configuration (I, S or SI) and on the process under study.

- Together with Ahn et al. (2023), we also found that there is no best performing BC method. Table 6 and Table 7 respec-
  tively summarize the results obtained in this study and provide recommendations reported above.





- Galmarini et al. (2024) considered a total of 12 crop models, which are highly multivariate and integrative in time, but the spatial dimension was not considered at all. They found that R2D2 (I.R2D2 configuration) was among the best performing method, which is in contradiction with our findings showing that I.R2D2 and SI.R2D2 do not adjust better than CDF-t for SWC and FWI. The exact reason for this discrepancy should be explored in future work.

- Ahn et al. (2023) explored MBC methods on hydrological models using only three input climate variables: minimum and maximum daily temperatures and daily precipitation. MBC methods were explored in the same three configurations (I, S, SI) as our study. Notice that the intervariable setting involves thus much less variables than ours and that hydrological models are highly integrative in time and space. They found that dOTC faces difficulties with increasing number of dimensions, generating deterioration in univariate correction. This was also reported in François et al. (2020) and it was visible in our study on FLO (which depends only in Temperature). However, SI.dOTC was found here to be the best BC method when the spatial component is important for highly multivariate and time integrative processes, such as SWC and FWI.

For time and computation considerations, we considered a single climate model (IPSL-CM6A-LR) in this work, and our analysis was limited to three regions in France. Our findings should be further confirmed in other parts of the world and using a larger set of climate models, possibly with higher resolution than the low-resolution IPSL-CM6A-LR model, which was available at that time.

In the present study, three indicators (Phenological stage FLO, Soil Water Content and Fire Weather Index) depend on temporal properties. However, none of the bias correction methods applied here were designed to adjust these properties. A few studies have developed BC methods for the adjustments of (some) temporal properties of climate variables in addition to inter-variable and spatial properties (e.g., Mehrotra and Sharma, 2019; Vrac and Thao, 2020; Robin and Vrac, 2021). However, the adjustment of temporal properties necessarily leads to modify, at best slightly, univariate, inter-variable and/or spatial properties. Including such bias correction methods in the present intercomparison study would provide an even more comprehensive overview of the performance of bias correction methods for impact studies.

Finally, a particular effort has been made in this study to explain the performances of correction methods according to the different statistical characteristics involved in the calculation of impact indicators. However, statistical characteristics may contribute in different ways to the values of impact indicators. For example, as already pointed out, values of SWC and FWI depend (potentially differently) on marginal, inter-variable and temporal properties of the same input climate variables. Quantifying the contribution of the different statistical characteristics to the variability of the impact indicators, for example using variance-based sensitivity analysis methods, would not only allow us to improve the understanding of the performances of bias correction methods, but also to target the simulated properties that need to be corrected and ultimately to provide a better guidance on how to apply them for impact studies.





| | | CDF-t | I.R2D2 | I.dOTC | S.R2D2 | S.dOTC | SI.R2D2 | SI.dOTC |
|---|---|---|---|---|---|---|---|---|
| Correct° | Univariate w. Time dep. (FLO) | ∼ | (∼) | (✗) | (∼) | (∼) | (∼) | (✗) |
| of 1d-stat. | Multivariate dep. (ET0) | ∼ | ✓ | ✓ | ∼ | ∼ | ✓ | ✓ |
| | Time and multivariate dep. (SWC, FWI) | ∼ | ✗ | ✓ | ∼ | ∼ | ✗ | ✓ |
| Correct° | Univariate w. Time dep. (FLO) | ∼ | (∼) | (✗) | (∼) | (∼) | (∼) | (✗) |
| of spatial | Multivariate dep. (ET0) | ∼ | ✗ | ∼ | ∼ | ∼ | ✓ | ✓ |
| struct. | Time and multivariate dep. (SWC, FWI) | ∼ | ✗ | ∼ | ∼ | ∼ | ✗ | ✓ |

**Table 6.** Summary of results for the multivariate BC methods to adjust univariate statistics and spatial structures of the different impact outputs. Note that these results are based on the adjustments of simulated outputs from the IPSL-CM6A-LR model. Green checks and red crosses indicate whether BC methods performed well for the metrics and the different statistical characteristics in rows. Orange tildes indicate cases for which the added value of BC has been found to be limited in our study. Parentheses indicate that a MBC is not useful, because the process is univariate.

| Method | Recommendation for multivariate processes (univariate processes should be corrected with a univariate method) |
|---|---|
| CDF-t | Fair but not always optimal corrections for all metrics in all situations. No gain / no risk option |
| I.R2D2 | Restricted to processes with no time integration and no spatial analysis |
| I.dOTC | Fairly good corrections of the marginals |
| S.R2D2 | Comparable to CDF-t in most cases |
| S.dOTC | Comparable to CDF-t in most cases |
| SI.R2D2 | Restricted to processes with no time integration. On those, better than S.R2D2 on spatial metrics |
| SI.dOTC | Outperforms CDF-t for multivariate processes with time integration, both on marginals and on spatial metrics |

**Table 7.** Summary of recommendations for BC methods to adjust muultivariate statistics and spatial structures of the different impact outputs.



**Supplementary material**

For size consideration, the comprehensive set of Figures and Tables is accessible in the Technical Report Allard et al. (2023)
which is available on-line at https://hal.inrae.fr/hal-04227826.

**Acknowledgments**

This work is part of the COMPROMISE project funded by the metaprogram Adaptation of Agriculture and Forest to Climate Change (AAFCC) of the French National Research Institute for Agriculture, Food & Environment (INRAE). MV acknowledges support from the "COESION" project funded by the French National program LEFE (Les Enveloppes Fluides et
l'Environnement). MV's work also benefited from state aid managed by the National Research Agency under France 2030 bearing the reference ANR-22-EXTR-0005 (TRACCS-PC4-EXTENDING project).

**Author's contributions**

DA had the initial idea and lead the study. DA, MV and IGCA designed the experiments and DA, MV and BF defined the statistical analyses. BF computed the bias corrected climate variables, IGCA provided the code for the plant models. DA
computed all process outputs, wrote the codes for the analyses and to plot the figures. All authors contributed to the analyses. DA wrote the first draft of the article with inputs from all co-authors.

**Competing interests**

The authors declare that no competing interests are present.

**Data availability**

The IPSL-CM6A-LR model data simulations as part of the CMIP6 climate model simulations can be downloaded through the Earth System Grid Federation portals. Instructions to access the data are available at: https://esgf-node.ipsl.upmc.fr/projects/cmip6-ipsl. The SAFRAN reanalysis dataset is available upon request to the French National Centre for Meteorological Research (CNRM, Météo-France CNRS). All computed indicators, for all models, regions and periods considered in this work are available at https://doi.org/10.57745/TUIHKT (Allard et al., 2024).



## Appendix: list of abbreviations

| | |
|---|---|
| GCM | General Circulation Model |
| RCM | Regional Circulation Model |
| CMIP6 | 6th Coupled Model Intercomparison Project |
| IPSL | Institut Pierre-Simon Laplace (here, IPSL-CM6A-LR model) |
| SAFRAN | French reanalysis system |
| | |
| ET0 | Reference evapotranspiration (in mm) |
| FLO | Flowering (in day of the year) |
| FWI | Fire Weather Index (no unit) |
| SWC | Soil Water content (in %) |
| ESS | Effective Sample Size |
| | |
| BC | Bias Correction |
| MBC | Multivariate Bias Correction |
| CDF | Cumulative Distribution Function |
| CDF-t | Cumulative Distribution Function – Transform |
| R2D2 | Rank Resampling for Distributions and Dependencies (an MBC method) |
| dOTC | dynamical Optimal Transport Correction (an MBC method) |
| I.R2D2, I.dOTC | Intervariable setting for R2D2 and dOTC, repsectively |
| S.R2D2, S.dOTC | Spatial setting for R2D2 and dOTC, respectively |
| SI.R2D2, SI.dOTC | Spatial and Intervariable setting for R2D2 and dOTC, respectively |



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
