# Peer review of "Assessing multivariate bias corrections of climate simulations on various impact models under climate change"

_Hydrology and Earth System Sciences, 2024_

## Author Response (AR1)

Responses to referees' comments about the manuscript

**Assessing multivariate bias corrections of climate simulations on various impact models under climate change,**
by
*Denis Allard, Mathieu Vrac, Bastien François and Iñaki García de Cortázar-Atauri*

First of all, we express our gratitude to the two referees for their general positive appraisal and for valuable comments and feedback. We made efforts to incorporate all of their suggestions, and we believe that the manuscript has been enhanced as a result.

Below, we provide a detailed response to the reviewers' comments, with the comments presented in black and our responses in blue.
* * *
**Anonymous Referee #1**

**Introduction:**

The introduction would benefit from a more detailed and systematic literature review of the development of multivariate bias correction methods. Discussing the advantages and disadvantages of existing methods would provide valuable context and help highlight the contribution of this study.

We agree that we have provided a rather rapid overview of existing Multivariate Bias Correction (MBC) methods. However, it is not possible in this paper to provide a detailed description of all methods in this paper, which is not a review paper. We have thus decided to expand the paragraph presenting the MBC methods (lines 58-69) in the original manuscript. It now reads (lines 69-87):

"To adjust biases in multivariate dependencies, some multivariate bias correction (MBC) methods have been recently developed in the literature. The objective of MBC is fundamentally the same as that of univariate BC: transforming climate model simulations so that selected statistical features match those of a reference data set over the calibration period. The difference with univariate BCs lies in the fact that statistical features are not only univariate, but also multivariate such as inter-variable correlations or spatial copula structure. MBC methods can be grouped into three categories based on how they adjust climate simulations, (Vrac (2018); Robin et al. (2019); François et al. (2020): 1) most methods belong to the "marginal/dependence" category that gathers MBCs adjusting separately univariate distributions and dependence properties. For precipitation outputs of RCMs, Bárdossy and Pegram (2012) proposed a correction of the spatial structure using matrix recorrelation or sequential correlation based on copulas. For correcting spatiotemporal biases for multiple variables, Cannon (2018) proposed an image processing technique for multivariate and spatial bias correction. Sharma (2019) and Guo et al. (2019) proposed a modification of the ranks (obtained from the empirical quantiles) based on resampling for the former and shuffling for the latter. In the same spirit, but based on CDF-t for the univariate correction instead a simple quantile mapping, Vrac (2018) and Vrac and Thao (2020) proposed a "Rank Resampling for Distributions and Dependencies" bias correction (R2D2), which will be detailed later. In principle, these approaches are not limited in terms of the number of variables, grid points in space, and the time scale considered. Nahar et al. (2018) proposed an independent component analysis approach to correct biases at multiple locations conjointly, but this approach must be applied independently to each climate variable.

Nguyen et al. (2019) considered a bias correction approach for time series using the spectrum in the frequency domain, applicable to multivariate time series or to several grid cells. François et al. (2021) proposed Cycle-GAN, a cycle-consistent adversarial network for the adjustment of spatial dependence structures of climate model projections. 2) The "successive conditional" correction approach, that adjusts the different simulated variables successively and conditionally on the previously adjusted ones (e.g., Piani and Haerter, 2012; Dekens et al., 2017). 3) The "all-in-one"' category that consists of MBCs simultaneously adjusting univariate and multivariate properties of climate simulations. Pan et al. (2021) proposed to learn an adversarial neural network for spatially coherent corrections of daily precipitations, while Robin et al. (2019) use optimal transport theory for multivariate and spatially coherent bias corrections."

**Figure 1**

It would be helpful if the subtitles provided more detailed information about each panel. Additionally, labeling the subfigures with identifiers like (a), (b), (c) or numbers like 1, 2, 3 would improve readability and make it easier to reference specific parts of the figure in the text.

Done, thanks.

**Line 180**

Including a simple formula to show how the Soil Water Content (SWC) is calculated would enhance the reader's understanding of the methodology and the variables involved.

We added the following sentence describing how SWC is computed (now line 198) "by computing $SWC_t = SWC_{t-1} + (pr_t - k.ET0_t )/ d$, where d is the depth of the soil and k is a constant depending on the considered crop".

**Line 187:**

Please discuss the impact of the simplicity of the model or assumptions mentioned here. Elaborating on how this simplicity might affect the results or interpretations would strengthen the credibility of the study.

Multiplying the type of soils with possibly different depth and/or different soil water capacities would have significantly increased the experiment without significantly changing our results. We ran sensitivity tests in some settings. Even though some differences were visible in the SWC values, the conclusions in terms of bias correction methods remained unchanged (not shown). Actually, this result also holds when considering different wheat varieties for example. In order to keep the experimental design at a manageable size, we decided to run the full study for a single soil type and a single variety per crop. We added the following sentence (now line 206):

"Early sensitivity tests have shown that when changing these parameters, even though some differences were visible in the SWC values, the conclusions in terms of bias correction methods that will be reported later remained unchanged (not shown). In order to keep the experimental design at a manageable size, we decided to run the full study for a single soil type. This choice also avoids problems of soil heterogeneity in each region and allows us to catch as much climate effect as possible."

**Line 216:**

Under climate change, inter-variable correlations might change over time. How does the methodology account for potential changes in inter-variable correlations in future climate scenarios? Addressing this point would clarify the robustness of the bias correction methods when applied under changing climatic

With dOTC, a transport map between the multivariate density on the model and that on the reference on the calibration period is estimated. This transport map is then used to correct the model outputs in the projection period. The underlying hypothesis is that this transport map remains identical in the future, but it does not assume that the inter-variable correlations do not change in the future. Indeed, dOTC accounts for the changes in the multivariate distribution (and thus in the inter-variable correlations) and incorporates them into the corrections (Robin et al., 2019). Regarding R2D2, the marginal distributions can change in the future while the copula is considered stationary in time, and identical to that in the reference. This might seem a strong assumption, but Vrac et al. (2023) showed that this was a conservative hypothesis preventing climate models from exhibiting over-varying inter-variable correlations, not corresponding to actual (historical) evolutions (Vrac et al., 2022). In conclusion, assuming a stationary copula corresponding to reference data is a safe choice.

We added the following text at the end of the presentation of the R2D2 method (line 239).

"While this may appear to be a strong assumption, Vrac et al. (2023) showed that it serves as a conservative hypothesis. This prevents climate models from exhibiting over-varying inter-variable correlations that do not correspond to actual (historical) evolutions (Vrac et al., 2022). Assuming a stationary copula corresponding to reference data is thus a safe choice."

Additional references:

Vrac, M., Thao, S., and Yiou, P.: Should multivariate bias corrections of climate simulations account for changes of rank correlation over time?, Journal of Geophysical Research: Atmospheres, 127, e2022JD036 562, 2022.

Vrac, M., Thao, S., and Yiou, P. (2023) : Changes in temperature–precipitation correlations over Europe: are climate models reliable?, Climate Dynamics, 60, 2713–2733, 2023.
* * *
**Referee #2**

The paper present a comparative analysis of multivariate bias correction methods, that is performed with great care, excellent methodological set up and wealth of evidence in support of the conclusions. As far as I am concerned the manuscript can proceed toward publication almost as it is, being also extremely well written and clear.

Thank you very much for this positive appraisal.

My 'almost' is motivated by two requests of clarification and eventually of modification.

1- the domain area selection: while one can appreciate the selection of highly diverse sub-region of France, one can also wonder why the areas are all comparable in size and what would happen to the analysis should these areas be much larger in size. Was the choice of these sizes motivated by computational constraints or the authors see a problem in the methods performance should the domain be larger? An elaboration on this aspects in the conclusions or the case set up, I assume, is due.

Very good point. We added the following in the discussion (now line 617):

"In this study, we have considered three areas of rather limited size containing between 259 and 337 grid points. CDF-t and Intervar (I) settings are insensitive to the size of the domain. R2D2 can in principle handle very large vectors, hence very large domains, but we have seen that the S and SI settings are not to be recommended when spatial coherence is important. On multivariate processes with time integration for which the spatial structure matters, SI.dOTC brings improvement in all tested situations, i.e. in areas ranging from 4000 to 5000 km$^2$. Further tests are necessary to assess SI.dOTC on a larger number of grid points."

2- From the Conclusions section:

"Galmarini et al. (2024) considered a total of 12 crop models, which are highly multivariate and integrative in time, but the spatial dimension was not considered at all. They found that R2D2 (I.R2D2 configuration) was among the best performing method, which is in contradiction with our findings showing that I.R2D2 and SI.R2D2 do not adjust better than CDF-t for SWC and FWI. The exact reason for this discrepancy should be explored in future work."

While the first part of this conclusion is indisputable, the second part merits a more nuanced language and detailed analysis. In fact in Table 3 of Galmarini et al. (2024) it clearly appears that while multivariate methods outperform univariate ones, CDFt outperforms all its siblings but also scores in the first second and third error rankings frequencies comparable to the other multivariate methods and also R2D2 (ranking 2 and 3). Therefore for the specific comparison with CDFt, Galmarini et al. (2024) conclusion is not that far from those of the current manuscript. Please modify accordingly, if you agree. We agree that CDFt appears to be an outstanding univariate method whose properties should be indeed further investigated.

Thank you for your comment. This is absolutely correct, we thus changed our text which now reads (lines 596 - 604).

"Galmarini et al. (2024) considered a total of 12 crop models, which are highly multivariate and integrative in time, but the spatial dimension was not considered at all. They found that R2D2 (I.R2D2 configuration) was among the best performing methods, second to CDF-t. Our results complement this finding with new results stemming from spatial settings. We also found that CDF-t is among the methods performing best, but that R2D2 underperforms in some situations. It must be remembered that R2D2 leaves the temporal dynamics of the pivot variable unchanged. Here, the pivot variable was temperature, and it was also the case in Galmarini et al. (2024). In this case, processes that are mostly driven by temperature will perform fairly well, as reported in Galmarini et al. (2024). When variables other than the pivot are important (i.e. for SWC and FWI which depend highly on precipitation), we found that I.R2D2 and SI.R2D2 do not adjust better than CDF-t for SWC and FWI."

As for the rest I have nothing else to object and while waiting for the response of the authors I wish to congratulate them on this nice piece of research.

Thank you very much once again.